# Womb to womb: Maternal litter size and birth weight but not adult characteristics predict early neonatal death of offspring in the common marmoset monkey

**Julienne N. Rutherford**[1]*, **Corinna N. Ross**[2], **Toni Ziegler**[3], **Larisa A. Burke**[4], **Alana D. Steffen**[5], **Aubrey Sills**[6], **Donna Layne Colon**[2], **Victoria A. deMartelly**[7], **Laren R. Narapareddy**[8], **Suzette D. Tardif**[2]

1 Department of Human Development Nursing Science, College of Nursing, University of Illinois Chicago, Chicago, Illinois, United States of America, 2 Southwest National Primate Research Center, Texas Biomedical Research Institute, San Antonio, Texas, United States of America, 3 Wisconsin National Primate Research Center, University of Wisconsin, Madison, Wisconsin, United States of America, 4 Office for Research Facilitation, College of Nursing, University of Illinois Chicago, Chicago, Illinois, United States of America, 5 Department of Population Health Nursing Science, College of Nursing, University of Illinois Chicago, Chicago, Illinois, United States of America, 6 Barshop Institute for Longevity and Aging Studies, University of Texas Health Science Center at San Antonio, San Antonio, Texas, United States of America, 7 Department of Biobehavioral Health Nursing Science, College of Nursing, University of Illinois Chicago, Chicago, Illinois, United States of America, 8 Nell Hodgson Woodruff School of Nursing, Emory University, Atlanta, Georgia, United States of America

* ruther4d@uic.edu

**Data Availability Statement:** The data for this paper are available from the OSF database, osf.io/sr7cg.

## Abstract

A singular focus on maternal health at the time of a pregnancy leaves much about perinatal mortality unexplained, especially when there is growing evidence for maternal early life effects. Further, lumping stillbirth and early neonatal death into a single category of perinatal mortality may obscure different causes and thus different avenues of screening and prevention. The common marmoset monkey (*Callithrix jacchus*), a litter-bearing nonhuman primate, is an ideal species in which to study the independent effects of a mother's early life and adult phenotypes on pregnancy outcomes. We tested two hypotheses in 59 marmoset pregnancies at the Southwest National Primate Research Center and the Barshop Institute for Longevity and Aging Studies. We explored 1) whether pregnancy outcomes were predicted independently by maternal adult weight versus maternal litter size and birth weight, and 2) whether stillbirth and early neonatal death were differentially predicted by maternal variables. No maternal characteristics predicted stillbirth and no maternal adult characteristics predicted early neonatal death. In univariate Poisson models, triplet-born females had a significantly increased rate of early neonatal death (IRR[se] = 3.00[1.29], p = 0.011), while higher birth weight females had a decreased rate (IRR[se] = 0.89[0.05], p = 0.039). In multivariate Poisson models, maternal litter size remained an independent predictor, explaining 13% of the variance in early neonatal death. We found that the later in the first week those neonates died, the more weight they lost. Together these findings suggest that triplet-born and low birth weight females have distinct developmental trajectories underlying greater rates of infant loss, losses that we suggest may be attributable to developmental disruption

**Funding:** JNR: Research reported in this publication was supported by the Eunice Kennedy Shriver National Institute of Child Health and Human Development of the National Institutes of Health (https://www.nichd.nih.gov/) under award number R01-HD076018 (Rutherford, PI). This investigation used resources that were supported by the Southwest National Primate Research Center grant P51-OD011133 from the Office of Research Infrastructure Programs, National Institutes of Health. The funders had no role in study design, data collection and analysis, decision to publish, or preparation of the manuscript.

**Competing interests:** The authors have declared that no competing interests exist.

of infant feeding and carrying. Our findings of early life contributions to adult pregnancy outcomes in the common marmoset disrupt mother-blaming narratives of pregnancy outcomes in humans. These narratives hold that the pregnant person is solely responsible for pregnancy outcomes and the health of their children, independent of socioecological factors, a moralistic framing that has shaped clinical pregnancy management. It is necessary to differentiate temporal trajectories and causes of perinatal loss and view them as embedded in external processes to develop screening, diagnostic, and treatment tools that consider the full arc of a mother's lived experience, from womb to womb and beyond.

## Introduction

Perinatal mortality, defined by the World Health Organization as encompassing stillbirth from the 22nd gestational week and early neonatal death up though the first postnatal week [1], is a common pregnancy outcome evaluated through the lens of maternal "risk factors." Deaths across this temporal range are often "grouped on the assumption that similar factors are associated with these losses [2, p. 178]." However, lumping stillbirth and early neonatal death into a single category may omit, combine, or obscure different causes [3, 4] and thus different avenues of screening and prevention. For example, maternal obesity is associated with a higher risk of stillbirth [5], neonatal mortality [6], and perinatal mortality [7], temporal periods with three different definitions. Different causes of death are possible given the different systems of dependence during these temporal periods. A fetus is dependent on the placenta, a direct physiological support system that is not controlled overtly by maternal behavior and provides buffering against extremes in nutrient availability and psychosocial stress [8]. A liveborn primate infant, on the other hand, is dependent both on a less physiologically-direct delivery of nutrients via breastmilk or other foods, and on a complex behavioral support system called "parenting" [9]. The differences in these delivery systems suggest that death in these contrasting temporal domains could have underlying causes that need to be differentiated.

Maternal health during gestation is thought to be a key feature underlying a successful pregnancy, one that ends in a live birth of a healthy infant. Maternal health is currently modeled as the amalgamation of a woman's genome, her current diet and lifestyle choices, her current weight or BMI, her current socioeconomic status, and her racial and/or ethnic characteristics, components frequently described as "risk factors" [10, 11]. However, there is a growing appreciation that adult health and thus maternal health surrounding a gestation is shaped not just by immediate biological and lifestyle characteristics but also by early life development, extending into the fetal period (e.g. "maternal ecology," [12]). Both animal and human studies show that fetal nutrient deprivation and/or growth disruptions are associated with altered reproductive parameters such as age of menarche/pubarche and age at first pregnancy, and outcomes such as offspring viability and birth weight, etc. [13, 14].

The common marmoset monkey (*Callithrix jacchus*) is an ideal nonhuman primate in which to study the consequences of variability in intrauterine environments and the independent effects of a mother's fetal/neonatal and adult phenotypes on her pregnancy outcomes. Marmosets produce litters ranging from twins to quadruplets in captivity, and larger litters experience differences in placentation, such as reduced surface area for nutrient transport [15] and poorer perinatal outcomes [16, 17]. Differences extend into adulthood. Marmosets born as small triplets are more likely to grow into large adults than are similarly small twins [18]. In a recent retrospective study, we found that triplet-born marmoset females, regardless of their

birth weight, produced the same number of offspring as their twin counterparts but lost three times as many to spontaneous abortion and stillbirth [19]. In our framing, litter size in the marmoset represents a range of intrauterine nutritive conditions, with triplets experiencing restrictive intrauterine environments compared to twins.

In the current study, we tested two hypotheses. The first hypothesis is that pregnancy outcomes are predicted independently by 1) maternal adult weight, with the prediction that perinatal mortality is higher when the mother is at extremes of weight and 2) maternal early life variables of litter size and birth weight, with the prediction that perinatal mortality is higher when the mother herself was born a triplet or at a lower weight. The second hypothesis is that the two categories of perinatal mortality, stillbirth and early neonatal death, may be related differently to the maternal predictors. This study differs from our previous work in that it comes from a prospective sample for which we have complete data about postnatal infant survival. The study differentiates two elements of perinatal mortality, stillbirth and neonatal death, to test the possibility that these outcomes follow different pathways through maternal biology across the lifecourse.

## Methods

### Ethics statement

All animal procedures, husbandry, and housing were approved by the Southwest National Primate Research Center (SNPRC) Institutional Animal Care and Use Committee requirements, certified by the SNPRC and the University of Illinois Chicago Animal Care Committee.

### Colonies and housing

Animals in this study all originated from the same founder population but were housed at two facilities, the SNPRC at the Texas Biomedical Research Institute and the Barshop Institute for Longevity and Aging Studies at the University of Texas Health Science Center at San Antonio (UTHSCSA), referred to throughout as the SNPRC and Barshop colonies. Diet, housing, husbandry, and handling procedures were the same at both sites though staff differed. Animals were housed in family groups in large metal wire enclosures, approximately one meter deep, two meters wide, and one meter tall, on a one meter tall metal frame. Enrichment was provided in the form of hanging and climbing structures and a nest box. All study females at both locations were housed with at least their adult male mate, but often with older offspring. It is not uncommon for the family group to contain adolescent, juvenile, and infant offspring at the same time. Family structure at the time of each pregnancy studied was not recorded and thus could not be considered in this study.

### Subjects

Fourteen adult female common marmoset monkeys were enrolled in this prospective study, seven at each colony. Barshop had five triplet-born females compared to three at SNPRC. Barshop produced 34 pregnancies compared to 25 at SNPRC. The females were all born between 2009 and 2012 and were of varied age and parity at the time of enrollment. Enrollment was initiated upon an ultrasound confirmation of pregnancy, with regular subsequent exams to track pregnancy, estimate gestational age based on fetal measurements, and predict delivery date [20]. These enrolled females are referred to as the F0 generation; their offspring are referred to as the F1 generation.

## Coding pregnancy loss and early neonatal death

Stillbirth of the F1 offspring was determined by direct observation at the time of birth and is reported as the number of individual fetuses out of a litter. A fetus was considered stillborn if it did not appear to be moving or breathing at the time of birth and if it was deemed by an experienced marmoset observer to be of appropriate size and development for gestational age at the time of delivery, with no sign of earlier death in utero; signs of death prior to expulsion include macerated flesh, skin slippage, and a general appearance of "mushiness" [21]. Spontaneous abortion (i.e., miscarriage) was determined when products of conception were observed in the cage prior to the expected due date, and/or subsequent ultrasound exams showed no evidence of a continued pregnancy. Since it is not possible to assign a definitive count of fetuses lost to abortion because of small size and unobserved fetophagy, abortion is coded as having occurred or not. All losses coded as abortions were of fetuses in approximately the first half of gestation. Losses in the second half of gestation, prior to labor and the peripartum, are rare and were not observed during this study. Early neonatal death of the F1 offspring was defined as death within seven days of birth. Review of individual records was performed to examine observations related to birth and death of a subset of neonates.

## Outcome and predictor variables

The outcomes of interest were counts of stillbirths and early neonatal death per litter by day seven. Day seven was chosen because it is similar to common definitions of human early neonatal mortality, with roughly 75% of human neonatal mortality [22] and 90% of marmoset neonatal mortality [23] occurring in this time frame. F0 predictors include maternal early life predictors (i.e., maternal litter size at birth (twin-born or triplet-born) and maternal birth weight) and maternal gestational predictors (i.e., weight at gestational day 60, day 90, and day 120 [of an approximately 143-day gestation] and total weight gain). Lunn (1983) demonstrated in the marmoset that weight at d. 60 of gestation is highly correlated with and not substantially greater than pre-pregnant weight [24], thus in this study it is used as a proxy for prepregnancy weight and to calculate total weight gain during pregnancy. Our earlier work [19] suggested sex ratio at birth might have implications for later reproductive success so we considered the relationship of the proportion of female littermates in a female's birth litter to other variables. Colony was included as a predictor to account for the disparity in sampling of triplet-born females at the two colonies as well as site-specific patterns arising from environmental or husbandry factors not otherwise captured. F1 predictors include litter size, total litter weight, and average neonatal weight (total weight/litter size).

## Statistics

The unit of analysis of the study was the pregnancy, not the individual female. We have shown previously that 1) individual female repeatability of litter size in these marmosets is low, 2) the litter size a female is born into does not predict the litter size she will produce as an adult, i.e., triplet females are not more likely to produce triplet litters, and 3) litter size is predicted by maternal mass prior to pregnancy, together suggesting that litter size is not genetically constrained but energetically flexible within and between individuals [16, 20, 21].

All analyses were conducted using Stata SE v.16.1 for Windows (StataCorp, College Station, TX). Independent samples two-tailed t-tests were used to compare F0 twins and triplets to each other in terms of birth condition and other life history characteristics. Significance was set as $\alpha \leq 0.05$. Cohen's d was calculated to report effect size [25]; a value of d = 0.2 is considered a small effect size, 0.5 a medium effect size and 0.8 and above a large effect size.

To explore relationships among the predictors in order to construct models, Pearson's pair-wise correlations were performed without correction for multiple comparisons. Because the dependent variables were counts of perinatal deaths, Poisson-adjusted models with offsets were used. For stillbirth Poisson models, litter size was defined as the offset. For early neonatal death models, the offset was defined as number of liveborn. Results were reported as incident rate ratios (IRR). Maternal ID was modeled as a random effect to account for multiple pregnancies. Initially relationships between predictors and offspring outcomes were explored via Poisson single predictor models. If multiple variables emerged as significant predictors of perinatal mortality categories, Poisson multiple regression models were constructed. Chi$^2$ was used to compare pregnancy counts per maternal litter size between the colonies and to compare birth status and cause of death for subset of neonates dying within one week in the Barshop colony.

## Results

### Reproductive demographics of the F0 adult females and their F1 pregnancies

The 14 F0 females produced a total of 59 pregnancies (Fig 1). Twelve (20.34%) ended in spontaneous abortion and 47 (79.66%) ended at term. Of those term pregnancies, one (2.13%) ended with the entire litter being lost to peripartum stillbirth, 29 (65.96%) with the entire litter being liveborn, and the remaining 15 (31.91%) resulting in a mix of live and stillborn infants. A total of 140 term F1 infants (liveborn + stillborn) were produced from those 47 term pregnancies, with litters ranging from 1 to 5 (Fig 1); more than half of all litters were triplet litters, producing more than half of all offspring, followed by quadruplets and then twins. Singletons and quintuplets were rare.

Not all data were available for all F0 females and pregnancies so individual sample sizes are reported for each cell in Table 1. Maternal birth weight and percentage of female littermates are reported as available for the 14 females in the study. The rest of the data are reported as available for the pregnancies in the study. For pregnancy-specific characteristics (i.e., dam age, gestational weight and weight gain, litter size, litter weight, stillbirths, neonatal losses) only results from term pregnancies are reported. Of 59 pregnancies, 26 births (44.07%), all term, were directly observed and form the basis for analysis of stillbirths and early neonatal deaths.

### Maternal litter size differences in early life and gestational characteristics

Six of the 14 F0 females were twin-born; eight were triplet-born. Birth weights were available for four of the twin-born females and seven of the triplet-born females; triplet-born females weighed less at birth than twin-born females though the difference was not statistically significant (Cohen's d = 1.26, p = 0.08; Table 1). Triplet-born females had 20% more female littermates than twin-born females, but the difference was not statistically significant (Cohen's d = -0.63, p = 0.27; Table 1).

Twin-born and triplet-born females did not differ significantly in adult weight at any time in gestation. Total gestational weight gain for twin-born females was 22.11% more than for triplet-born during gestation (Cohen's d = 0.45), but the difference was not significant (p = 0.21). Twin-born and triplet-born females did not differ significantly in the size of litters they produced (2.82 vs. 3.32 infants, respectively; Cohen's d = 0.45; p = 0.11). Triplet-born females produced litters that weighed less than did twin-born (72.24 g vs. 96.92 g, respectively; Cohen's d = 0.95, p = 0.003). The average weight of individual infants of triplet-born females was 11.11% less than that of infants of twin-born; the difference was not statistically significant

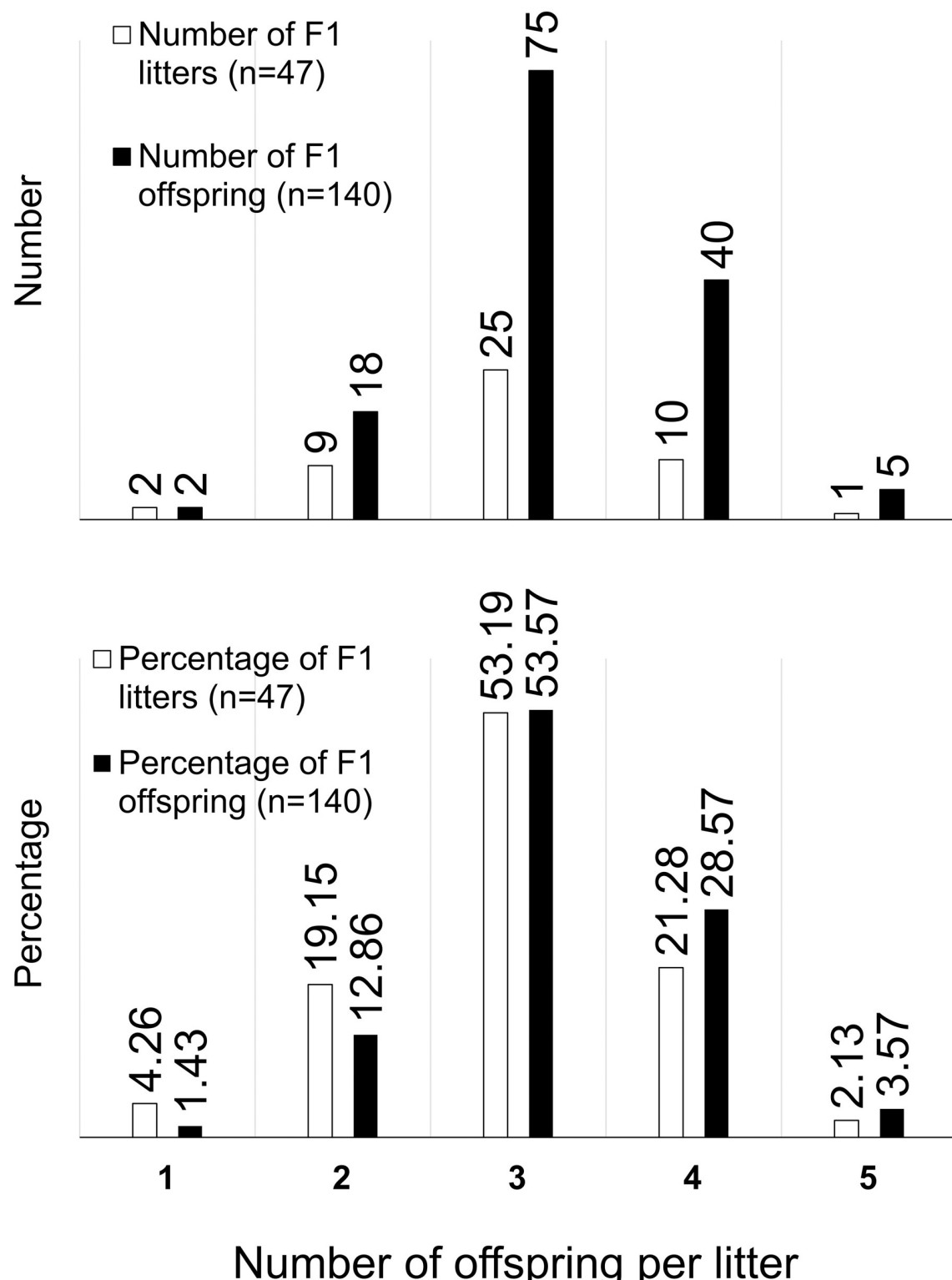

**Fig 1. Range of litter sizes in absolute numbers and percentages.** Triplet litters are most common, producing the greatest number of offspring overall.

**Table 1. Descriptive statistics and differences in pregnancy characteristics by maternal litter size.**

| | Combined Mean (±SD) | Twin-born F0 Mean (±SD) | Triplet-born F0 Mean (±SD) | *P* value* | Cohen's d |
|---|---|---|---|---|---|
| F0 birth weight, g | n = 11 | n = 4 | n = 7 | 0.08 | 1.26 |
| | 32.15 | 35.00 | 30.51 | | |
| | (4.07) | (3.72) | (3.48) | | |
| F0 female littermates, % | n = 14 | n = 6 | n = 8 | 0.27 | -0.63 |
| | 55.95 | 50.0 | 60.42 | | |
| | (16.80) | (0) | (21.71) | | |
| F0 age at time of pregnancy, years | n = 59 | n = 22 | n = 37 | 0.22 | -0.34 |
| | 4.37 | 4.20 | 4.48 | | |
| | (0.11) | (0.74) | (0.90) | | |
| Pregnancies in study | n = 59 | n = 22 | n = 37 | 0.44 | -0.21 |
| | 2.93 | 2.73 | 3.05 | | |
| | (1.56) | (1.45) | (1.63) | | |
| 60 d. gestational weight, g | n = 47 | n = 19 | n = 28 | 0.63 | 0.14 |
| | 456.79 | 464.35 | 451.66 | | |
| | (87.79) | (104.02) | (76.48) | | |
| 90 d. gestational weight, g | n = 42 | n = 17 | n = 25 | 0.95 | 0.02 |
| | 465.70 | 466.73 | 464.99 | | |
| | (85.21) | (106.14) | (69.95) | | |
| 120 d. gestational weight, g | n = 43 | n = 18 | n = 25 | 0.95 | 0.06 |
| | 512.43 | 511.50 | 513.10 | | |
| | (80.23) | (100.62) | (63.97) | | |
| Gestational weight gain, g | n = 33 | n = 15 | n = 18 | 0.21 | 0.45 |
| | 64.35 | 71.40 | 58.47 | | |
| | (29.06) | (26.14) | (30.77) | | |
| F1 litter size** | n = 47 | n = 20 | n = 27 | 0.11 | 0.43 |
| | 2.98 | 3.2 | 2.82 | | |
| | (0.82) | (0.83) | (0.79) | | |
| F1 litter weight, g | n = 45 | n = 19 | n = 26 | ***0.003*** | 0.95 |
| | 82.05 | 94.78 | 72.74 | | |
| | (25.40) | (25.87) | (20.98) | | |
| F1 average fetal weight, g | n = 46 | n = 20 | n = 26 | 0.10 | 0.54 |
| | 27.47 | 28.80 | 26.44 | | |
| | (4.80) | (4.43) | (4.91) | | |
| F1 litters lost to spontaneous abortion | n = 59 | n = 22 | n = 37 | 0.10 | -0.45 |
| | 0.20 | 0.09 | 0.27 | | |
| | (0.41) | (0.29) | (0.45) | | |
| F1 offspring lost to stillbirth*** | n = 24 | n = 11 | n = 13 | 0.57 | 0.22 |
| | 0.29 | 0.36 | 0.23 | | |
| | (0.55) | (0.67) | (0.44) | | |
| F1 offspring lost to early neonatal death | n = 43 | n = 19 | n = 24 | ***0.004*** | -0.95 |
| | 1.12 | 0.63 | 1.50 | | |
| | (1.05) | (0.83) | (0.98) | | |

* Independent two-tailed T-test, α≤0.05.

** Term neonates only, excludes aborted fetuses; Number of litters multiplied by litter size = total number of term offspring in study.

*** Observed births only.

(Cohen's d = 0.54, p = 0.10). Triplet-born did not differ from twin-born in their average age at time of pregnancy (Cohen's d = -0.34, p = 0.22) or number of pregnancies in the study (Cohen's d = -0.21, p = 0.44).

## Maternal litter size differences in pregnancy loss categories

Overall, triplet-born mothers lost 2.38 times more neonates in the first week of life than did twin-born (Cohen's d = -0.95, p = 0.004; Table 1). They also had 3 times more spontaneous abortions, but this difference was not significant (Cohen's d = -0.45, p = 0.10). The two groups did not differ in number of stillbirths (Cohen's d = 0.22, p = 0.57).

## Perinatal mortality predictors

**Correlations among predictors.** In correlation analyses (Table 2), maternal birth weight and litter size are treated as the primary early life predictors of individual term pregnancies that led to directly observed births (n = 26), rather than descriptors of individual F0 mothers

**Table 2. Correlation matrix of predictor variables.**

| | Colony | F0 birth weight△ | F0 litter size△ | % female F0 litter | F0 age at preg. | d. 60 weight | d. 90 weight | d. 120 weight | Weight gain | F1 litter size | Total F1 Litter weight | Mean F1 neonate weight |
|---|---|---|---|---|---|---|---|---|---|---|---|---|
| **F0 birth weight△** | n = 20 r = 0.12 | - - | - - | - - | - - | - - | - - | - - | - - | - - | - - | - - |
| **F0 litter size△** | n = 26 r = -0.30 | **n = 20 r = -0.75***** | - - | - - | - - | - - | - - | - - | - - | - - | - - | - - |
| **% female, F0 litter** | n = 26 r = 0.12 | **n = 20 r = -0.47**** | n = 26 r = 0.30 | - - | - - | - - | - - | - - | - - | - - | - - | - - |
| **F0 age at preg.** | n = 26 r = 0.05 | n = 20 r = -0.09 | n = 26 r = -0.05 | **n = 26 r = 0.45**** | - - | - - | - - | - - | - - | - - | - - | - - |
| **d. 60 weight** | n = 21 r = 0.34 | n = 16 r = 0.34 | n = 21 r = -0.18 | n = 21 r = -0.24 | n = 21 r = -0.27 | - - | - - | - - | - - | - - | - - | - - |
| **d. 90 weight** | n = 23 r = 0.24 | n = 17 r = 0.32 | n = 23 r = -0.11 | n = 23 r = -0.25 | n = 23 r = -0.27 | **n = 21 r = 0.97********* | - - | - - | - - | - - | - - | - - |
| **d. 120 weight** | n = 23 r = 0.25 | n = 17 r = 0.36 | n = 23 r = -0.14 | n = 23 r = -0.11 | n = 23 r = -0.21 | **n = 21 r = 0.93********* | **n = 23 r = -0.97********* | - - | - - | - - | - - | - - |
| **Weight gain** | n = 16 r = 0.17 | n = 11 r = 0.43 | n = 16 r = -0.29 | n = 16 r = 0.21 | n = 16 r = 0.18 | n = 16 r = 0.06 | n = 16 r = 0.24 | n = 16 r = 0.44* | - - | - - | - - | - - |
| **F1 litter size** | n = 26 r = 0.26 | n = 20 r = 0.06 | n = 26 r = -0.18 | n = 26 r = -0.10 | n = 26 r = -0.17 | n = 21 r = 0.14 | n = 23 r = 0.18 | n = 23 r = 0.23 | n = 16 r = 0.33 | - - | - - | - - |
| **Total F1 Litter weight** | n = 24 r = 0.21 | **n = 18 r = 0.63***** | **n = 24 r = -0.46**** | n = 24 r = -0.18 | n = 24 r = 0.08 | n = 21 r = 0.35 | n = 23 r = 0.38* | **n = 23 r = 0.44**** | **n = 16 r = 0.54**** | **n = 24 r = 0.79********* | - - | - - |
| **Mean F1 neonate weight** | n = 25 r = -0.06 | **n = 19 r = 0.57***** | n = 25 r = -0.27 | n = 25 r = -0.08 | n = 25 r = 0.35* | n = 21 r = 0.32 | n = 23 r = 0.33 | n = 23 r = 0.33 | n = 16 r = 0.31 | n = 25 r = -0.29 | **n = 24 r = 0.44**** | - - |
| **% female F1 litter** | n = 25 r = 0.02 | n = 19 r = 0.07 | n = 25 r = -0.19 | n = 25 r = -0.34* | n = 25 r = -0.27 | n = 21 r = 0.04 | n = 23 r = 0.09 | n = 23 r = 0.06 | n = 16 r = 0.27 | n = 25 r = 0.28 | n = 24 r = 0.34 | n = 25 r = 0.17 |

▲ Matrix includes only directly observed births.

⌂ Maternal early life predictors.

*p≤0.10.

**p≤0.05.

***p≤0.01.

****p≤0.0001.

**Table 3. Poisson univariate models\* predicting stillbirth and neonatal death.**

| Predictors | Stillbirth\*\* | | | | | Neonatal death within 1 week\*\*\* | | | |
| --- | --- | --- | --- | --- | --- | --- | --- | --- | --- |
| | unit increase | Observation n (Dam n) | IRR | SE | p-value | Observation n (Dam n) | IRR | SE | p-value |
| Maternal characteristics | | | | | | | | | |
| Colony (Barshop) | 1 | 24 (13) | 1.87 | 1.66 | 0.479 | 23 (12) | 0.44 | 0.18 | 0.054 |
| **Dam birth weight** | 1 | 18 (10) | 1.11 | 0.17 | 0.483 | 18 (10) | 0.89 | 0.05 | **0.039** |
| **Dam litter size of 3 (vs 2)** | 1 | 24 (13) | 0.70 | 0.60 | 0.678 | 24 (13) | 3.00 | 1.29 | **0.011** |
| Percent female in dam litter | 1 | 24 (13) | 0.97 | 0.04 | 0.454 | 24 (13) | 1.04 | 0.12 | 0.747 |
| Dam age at delivery (year) | 1 | 24 (13) | 0.93 | 0.52 | 0.901 | 24 (13) | 0.85 | 0.21 | 0.513 |
| 60 d. gestational weight (g) | 1 | 21 (11) | 1.00 | 0.01 | 0.472 | 21 (11) | 1.00 | 0.00 | 0.793 |
| 90 d. gestational weight (g) | 1 | 23 (12) | 1.00 | 0.00 | 0.470 | 23 (12) | 1.00 | 0.00 | 0.782 |
| 120 d. gestational weight (g) | 1 | 23 (12) | 1.00 | 0.01 | 0.410 | 23 (12) | 1.00 | 0.00 | 0.968 |
| Total weight gain (g) | 1 | 16 (9) | 1.00 | 0.02 | 0.791 | 16 (9) | 0.99 | 0.01 | 0.068 |
| Litter characteristics | | | | | | | | | |
| Litter weight (g) | 10 | 23 (12) | 1.01 | 0.02 | 0.665 | 23 (12) | 0.99 | 0.01 | 0.325 |
| Average infant weight (g) | 1 | 24 (13) | 0.90 | 0.07 | 0.155 | 24 (13) | 0.97 | 0.03 | 0.304 |
| F1 litter size | 1 | 24 (13) | 2.41 | 1.31 | 0.106 | 24 (13) | 0.98 | 0.26 | 0.927 |
| Percent female in F1 litter | 10 | 24 (13) | 1.00 | 0.01 | 0.957 | 24 (13) | 0.99 | 0.01 | 0.167 |

\* Random Effect: dam ID.

\*\*Litter size is the offset.

\*\*\*Number of livebirths is the offset.

(n = 14), to form the rationale for subsequent modeling. As expected, the maternal early life characteristics of maternal litter size and maternal birth weight were significantly inversely correlated across pregnancies (n = 20, r = -0.75, p<0.00001). Maternal litter size was also significantly inversely correlated with offspring litter weight (n = 24, r = -0.46, p = 0.02); triplet-born females produce lighter litters (Table 1). Maternal birth weight was significantly positively correlated with total offspring litter weight (n = 18, r = 0.63, p = 0.01) and average neonate weight (n = 19, r = 0.57, p = 0.01).

Our earlier work [19] suggested sex ratio at birth might have implications for later reproductive success, so we considered the relationship of the percentage of female littermates in a female's birth litter to other predictor variables. Only maternal age was correlated with female ratio such that older females came from more female-dominant litters (n = 26, r = 0.45, p = 0.02).

Gestational weights at three time points were all significantly correlated with each other. Larger females stayed larger; smaller females stayed smaller. Gestational weight at day 120 as well as total weight gain were significantly correlated with total litter weight.

**Predictors of stillbirth.** In Poisson single predictor models with total F1 litter size as the offset and maternal ID as a random effect, number of stillbirths was not predicted by any maternal characteristic (Table 3). Larger litters were substantially more susceptible to stillbirth (IRR[se] = 2.41[1.31]) though this was not significant (p = 0.106).

**Predictors of early neonatal death.** Maternal litter size and maternal birth weight were the only significant independent predictors of number of neonates dying in the first week of life (Table 3). Triplet-born mothers had a significantly increased rate of early neonatal death (IRR[se] = 3.00[1.29], p = 0.011), while mothers with higher birth weights had a decreased rate (IRR[se] = 0.89[0.05], p = 0.039). Living in the SNPRC colony, the colony with fewer triplet-

**Table 4. Poisson multivariate regression models* predicting neonatal death**.**

| Characteristic | IRR | SE | LCI | UCI | p-value | VIF | Pseudo-R$^2$ |
|---|---|---|---|---|---|---|---|
| Model 1: Colony, dam birth weight, dam litter size (n = 18) | | | | | | | 0.12 |
| Colony | 0.74 | 0.35 | 0.29 | 1.86 | 0.516 | 1.07 | |
| Dam Birth Weight | 0.95 | 0.09 | 0.80 | 1.14 | 0.583 | 2.16 | |
| Dam Litter Size | 1.76 | 1.31 | 0.42 | 7.53 | 0.430 | 2.11 | |
| *Overall model* | | | | | 0.157 | | |
| Model 2: Colony, dam birth weight (n = 18) | | | | | | | 0.11 |
| Colony | 0.71 | 0.33 | 0.28 | 1.76 | 0.456 | 1.07 | |
| Dam Birth Weight | 0.90 | 0.05 | 0.80 | 1.01 | 0.080 | 1.07 | |
| *Overall model* | | | | | 0.095 | | |
| Model 3: Colony, dam litter size (n = 24) | | | | | | | 0.13 |
| Colony | 0.67 | 0.30 | 0.28 | 1.61 | 0.367 | 1.20 | |
| Dam Litter Size | 2.52 | 1.18 | 1.00 | 6.32 | ***0.049*** | 1.20 | |
| *Overall model* | | | | | ***0.027*** | | |
| Model 4: Dam birth weight, dam litter size (n = 18) | | | | | | | 0.11 |
| Dam Birth Weight | 0.94 | 0.08 | 0.80 | 1.12 | 0.497 | 2.11 | |
| Dam Litter Size | 1.86 | 1.33 | 0.46 | 7.54 | 0.384 | 2.11 | |
| *Overall model* | | | | | 0.090 | | |

*Random Effect: dam ID.

**Offset: number of liveborn offspring.

born females, predicted a lower rate ratio of early neonatal death (IRR[se] = 0.44[0.18]) though this association was not significant (p = 0.054).

Four Poisson multivariate models of the two significant maternal early life variables plus colony as predictors were constructed (Table 4), one including all three predictors (Model 1), one including colony and maternal birth weight (Model 2), one including colony and maternal litter size (Model 3), and one including maternal birth weight and maternal litter size (Model 4). Only Model 3, colony and maternal litter size, was significant overall, and in that model, only maternal litter size remained a significant predictor of early neonatal death. This model accounted for 13% of the variance in early neonatal mortality.

## Neonatal characteristics contributing to early death

We examined the timing and apparent cause of death of a subset of 34 neonates, all from the Barshop colony, who died within the first postnatal week (Table 5). All deaths within the first two days were associated with observed weakness at birth (e.g. having difficulty grasping and

**Table 5. Neonatal deaths within first week of life.**

| Day of death | N | Weak at birth | | | Deaths associated with trauma | | | Average percentage of birth weight lost |
|---|---|---|---|---|---|---|---|---|
| Day 1 | 13 | 13 | 100% | Chi$^2$(1) = 33.00 ***p<0.0001*** | 6 | 88.9% | Chi$^2$(1) = 2.75 ***p<0.0001*** | 0%* |
| Day 2 | 10 | 10 | | | 2 | | | 4%* |
| Day 3 | 6 | 0 | 0% | | 0 | 11.1% | | 9%* |
| Days 4–7 | 5 | 0 | | | 1 | | | 17%** |

*Within range of normal 10% weight loss of breastfed human infants.

** Outside range of normal 10% weight loss of breastfed human infants.

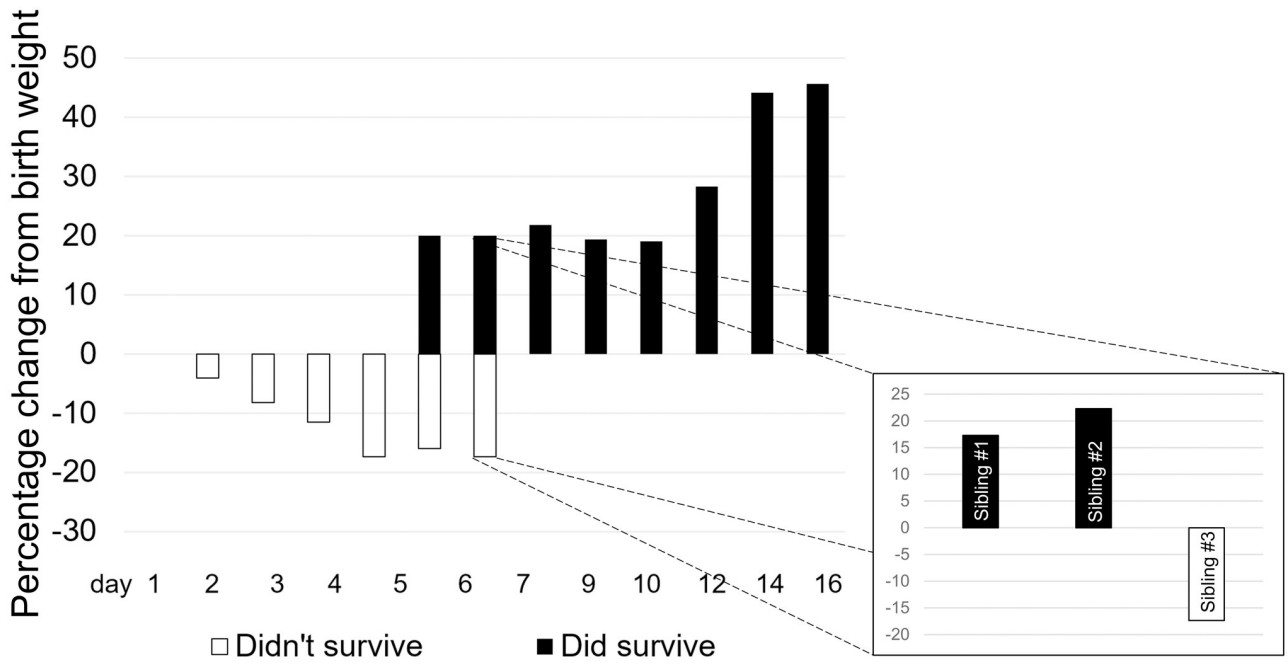

**Fig 2. Comparison of weight change in F1 infants who died in the first week (n = 34) and those who survived (n = 11).** Inset: comparison of weight change for one triplet litter with one loss and recorded weights for all siblings on day 7.

holding on to parent or sibling, including falling and overall locomotor difficulty). Deaths on days 1–2 were significantly more likely than those on days 3–7 to be associated with weakness at birth (Chi$^2$(1) = 33.00, p<0.0001) and trauma (such as a group-inflicted injury) prior to death (Chi$^2$(1) = 2.75, p<0.0001). Infants dying on days 3–7 did not show obvious signs of weakness at birth but declined afterwards; they were associated with weight loss approaching and then exceeding the 10% weight loss considered normal for exclusively breastfed human infants within the first week [26]. Weight change among a subset of surviving marmoset offspring in this study (n = 11) showed marked weight gain averaging 18.44% of birth weight through day 12 (Fig 2). Where there was temporal overlap, days 6–7, those who died (n = 2) lost an average of 16.7% of their birth weight in contrast to those who survived (n = 3), who gained 20% of their birth weight. In one case, all siblings were weighed when one was found dead on day 7 (Fig 2, inset). The dead sibling had lost 17.36% of its birth weight, while the two living siblings gained 17.28% and 22.48% respectively.

## Discussion

Our findings demonstrate that stillbirth and early neonatal death may be the consequences of different pathways, and that maternal early life characteristics play an important and clinically underappreciated role in perinatal loss, specifically early neonatal death. We found that 1) neither adult maternal gestational weight nor weight gain nor age predicted perinatal loss in either category, 2) mothers who were triplet-born or born at lower birth weights had a significantly increased rate of early neonatal death, and 3) early neonatal death had its own temporal pattern. Taken together our findings suggest that our understanding of perinatal mortality is complicated by intersections between the gestational present and the maternal past, including the size of the litter, the developmental stage of the offspring, whether the offspring are supported internally or externally, and maternal developmental experience (Fig 3).

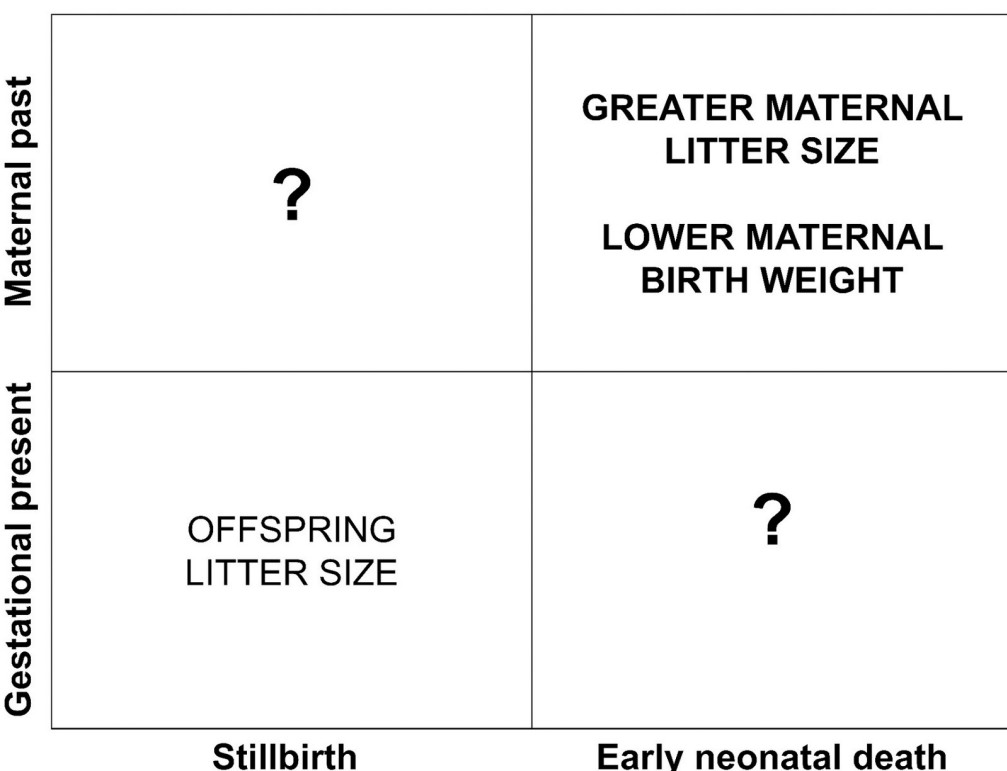

**Fig 3. The maternal past, operationalized in this study as the early life characteristics of maternal birth weight and litter size, appears to exert influence on the development of systems that support early neonatal survival.** The gestational present, in this case the size of the offspring litter, are more salient to the survival of the litter during gestation. The extent to which maternal past influences in utero survival, or the gestational present influences life outside the womb are unclear.

We hypothesized that heavier females would suffer greater perinatal loss in both categories, but we did not find that a proxy for prepregnant weight (i.e., day 60 weight), weight later in gestation, or total weight gain were associated with either category of perinatal mortality. The clinical literature finds a relationship between maternal obesity and increased perinatal mortality [27, 28], however the prevalence of maternal obesity was very low in this sample of marmosets. Only two females in our study could be categorized as obese (i.e., 90[th] percentile body weight [29]), limiting our ability to determine the relationship between obesity and pregnancy outcomes. We did show that the larger the litter being carried the more likely it was to be affected by stillbirth; though the difference was not significant, the rate of stillbirth for triplet-born mothers was 140% greater than for twin-born mothers. Interestingly, we have shown in the past that ovulation number and resultant litter size are associated with maternal mass [19, 30], suggesting that there may be an underlying link between maternal weight and stillbirth that we were not able to capture in this sample; we did not show a relationship between maternal adult weight and litter size here. It could also reflect that obesity is problematically cast as a "risk factor" during pregnancy [31] and that obesity itself may not be "risky" but rather a reflection of a cascade of developmental events and processes that themselves impact outcomes differentially. Recent analyses by the NICHD-established Stillbirth Collaborative Research Network found that models including maternal factors, including obesity, restricted to the immediate pregnancy explained only 19% of the variance in stillbirth outcomes, and nearly a third of stillbirths had no probable cause under current clinical paradigms [32, 33].

In contrast, early neonatal survival was predicted by maternal early life characteristics but not by present maternal gestational condition or aspects of the litter. Neither offspring litter size nor birth weight nor maternal weight at any time were associated with early neonatal mortality, consistent with previous studies [23]. On the other hand, mothers who were triplet-born and were born at lower weights had a significantly elevated rate of early neonatal mortality. Triplet-born mothers lost infants in the first week at a rate nearly 200% of that of twin-born. The rate of early neonatal death decreased nearly 10% with each gram increase in maternal birth weight. Our multivariate Poisson models explained 11–13% of the variance in early neonatal death in this sample. The one significant multivariate Poisson model in which only maternal litter size remained a significant predictor explained 13% of the variance, highlighting the potential value of considering maternal early life characteristics to better understand patterns of neonatal death.

What about maternal early life characteristics may influence infant survival? Early neonatal survival is dependent on several factors including the inherent robusticity (e.g., litter size, birth weight, motor skills) of the neonate itself and the quality of parenting it receives, especially carrying and feeding. The association of early neonatal death with early life maternal characteristics suggests that one potential underlying pathway could be the effect of developmental experiences on the physiological and behavioral underpinnings of parental characteristics. Variable rearing experiences may play an important role in the development of adult behaviors that influence neonatal viability. Studies in rodents and nonhuman primates have demonstrated that individuals separated from their mothers early in life, either intermittently or permanently, demonstrate increased stress reactivity in adulthood (e.g. rats, [34, 35]; primates, [36, 37]). For example, in rhesus macaques, females who themselves were raised in peer groups without mothers exhibited a high rate of premature infant rejection [38]. Quality of caretaking, not just its absence, also has an organizing effect on offspring behavior in adulthood. Classic studies by Michael Meaney and colleagues have elegantly demonstrated the lasting impact of caretaking behavior [39]. Rats who were reared by more attentive dams (i.e. higher levels of licking and grooming) had reduced hypothalamus-pituitary-adrenal (HPA) reactivity (e.g. lower serum levels of adrenocorticotrophic hormone [ACTH] and corticosterone]) as adults [40]. The daughters of less attentive dams were less attentive to their own offspring in turn [41].

This intergenerational pattern could play out for marmoset triplets. Marmoset mothers, unlike many other litter-bearing mammals, do not increase the total time invested in a litter as the litter increases in size: "the marmoset mother's tolerance for carrying infants does not increase with increasing infant stimuli present; rather, marmoset mothers will only tolerate a limited amount of time transporting and nursing infants, regardless of litter size [16, p. 825]." Triplet and lower birth weight infants were carried for shorter periods of time by their mothers; lower birth weight infants were nursed for shorter periods [16]. Mothers directed aggression to triplet and low birth weight infants at a higher rate than to twin and average weight infants. In sum, marmoset mothers of triplets and smaller infants may invest less time parenting their offspring, and that which is invested may be of lesser quality (e.g., more aggressive). A recent study showed large effect sizes of post-stressor cortisol concentrations in adult triplet-born marmosets who were either family-reared (where only two of the triplets survived) or human-supplemented compared to twin-born, family-reared marmosets; the sample was small and the findings were not significant but the evidence is "suggestive of differences in HPA axis activity between rearing conditions" for triplets [42]. It is thus possible that marmoset mothers who are themselves triplets or born at lower weights develop a less attentive maternal phenotype as a result of the parenting they were exposed to, in addition to the in utero exposures to nutritional stress, leading to a greater risk of neonatal loss, particularly during the

period of intense dependency on maternal and alloparental care and the availability and quality of milk.

Quantity and quality of milk are important factors shaping infant behavior and temperament [43]. Very little is known about how a mother's own developmental and metabolic experience, reaching back before her birth and extending to the birth of her own offspring, shapes the milk she produces, and sets into motion related consequences for her offspring [44]. Those consequences could involve lactational performance in adulthood. Quinn and colleagues have shown in humans that the quantity of milk epidermal growth factor and its receptor is related to maternal birth weight: women born at lower birth weights produced milk in adulthood with reduced levels of these bioactives [45]. Similarly, evidence from a pig model demonstrates that colostrum contains relaxin, a powerful organizer of female reproductive development and performance, and that relaxin is required for optimal infant cervical tissue development [46, 47]. Gilts who were deprived of colostrum as infants had reduced lifetime fecundity as adults, suggesting that early milk consumption plays an important role in developing the reproductive program [48]. These recent findings together further reinforce our womb to womb framing as it relates to milk composition and its consequences, particularly if smaller or triplet-born marmoset infants are fed less often than larger or twin-born infants.

Though we did not find an impact of neonatal litter size or weight on early mortality, we did show that deaths early in the first week were significantly associated with observed weakness and trauma, markers of reduced robusticity, and that later deaths were associated with losing weight in great excess compared to surviving marmosets. This pattern could be a reflection of lactation deficits in triplet-born and lower birth weight mothers, either in the latency to initiation, the amount of colostrum or milk produced, the quality of the milk produced, the quality of caretaking, or some combination thereof, all of which are associated with neonatal morbidity and mortality in humans [49, 50]. We did not determine the cause of death (with the exception of obviously fatal trauma) so infection, the leading cause of human early neonatal mortality globally [1, 22], cannot be ruled out. However, delayed initiation of breastfeeding is associated with increased risk of infection. Future studies to more closely monitor neonatal and (allo)parental behavior surrounding lactation and feeding, specific cause of death, and differences in maternal behavior among adult females are needed to fully assess the hypothesis that these later early deaths are linked to variation in lactation and parenting behavior as well as maternal developmental characteristics.

Frye et al. (2019) reported that *Callithrix jacchus* females from mixed sex litters exhibited impaired coordination on behavioral tests administered within 48 hours of birth [51]. This is of particular interest in unpacking our finding that maternal litter size was predictive of early neonatal mortality. In contrast to our earlier retrospective study [19], we did not find here an effect of sex ratio for either the mother or the litter on birth outcomes in the current study. However, triplet females are more likely to develop in mixed sex litters than are twins [19]. Marmosets are communal breeders with high levels of alloparental care, yet during the first week of life the mother is the primary caretaker [52, 53]. The extent to which successfully lactating and carrying infants during the first week is linked to maternal coordination is unknown, but this could be another developmental link to greater neonatal losses in that period for mothers who are triplets. Frye et al. did not test whether litter size was predictive of differences in coordination; previous studies of early marmoset development found that twins and triplets did not differ in how well first day behavioral scores predicted postnatal survival but these tests did not directly assess coordination [23]. More data on the duration of this difference in coordination through adulthood is needed to fully assess this hypothesis.

This is the first study of which we are aware that disentangles maternal pathways to stillbirth and early neonatal deaths in a nonhuman primate species. Our findings suggest that

efforts at lowering rates of perinatal mortality may be hampered when stillbirth and neonatal mortality are lumped together as the causes may have very different temporal and physiological roots. A few limitations remain in this prospective study regarding assessments of stillbirth and neonatal mortality. Given that fetophagy is not an uncommon occurrence and that parturition is typically nocturnal, our estimates of loss of even late gestation fetuses may be underestimated but because of the prospective nature of this study, every effort was made to be present at all study births; our use of serial sonography to track fetal number and growth suggests we did not miss any term offspring, even if we missed the moment of birth. Our observations of stillbirth are restricted to term losses and we performed statistics only on those stillbirths that could be confirmed at the time of birth (as opposed to finding a dead neonate the next morning); whether they were pre- or intrapartum losses we could not determine, and it is possible that dying in utero as opposed to dying during labor have different underlying causes. We did not conduct lung flotation, a highly accurate method of determining stillbirth in humans [54], but in the past when this has been conducted it was reflective of the described characteristics. Family composition was not considered so we do not know the extent to which being a first-time parent or simultaneously rearing older offspring might influence both stillbirth and early neonatal death. Age is typically correlated with parity and parental experience; previous work has found increases in infant mortality with maternal age [55] but we found no associations between maternal age (which ranged from 2.65–6.41 years) and any outcome variable in this study. We did not consider the specific impact a father's early life and adult characteristics might have on these outcomes [56]; those analyses are underway.

## Public health and health equity implications

Our findings of early life contributions to adult pregnancy outcomes in the common marmoset disrupt mother-blaming narratives of pregnancy outcomes in humans. These narratives hold that the pregnant person is solely responsible for pregnancy outcomes and the health of their children, independent of socioecological factors, a moralistic framing that has shaped clinical pregnancy management. A neoliberal emphasis on individualized management of health has largely localized the responsibility for "healthy" pregnancies to the pregnant person's body and its societally-constructed and clinically-defined "flaws" [31]; for example, shame is employed as a strategy to problematize the flaw of fatness during pregnancy [57]. Mother-blaming narratives are often written in racist ink, where erroneous biological notions of "race"–abstracted from the structural forces of racism in action across generations–are identified as an isolated and deterministic risk factor [58, 59]. Developmental Origins of Health and Disease (DOHaD) posits that intrauterine and early childhood developmental environments have lifelong implications; this approach too has the potential to concentrate blame on pregnant people by reducing them to the receptacle of that environment [60, 61]. While the fetus is of course inextricably linked to the intrauterine environment and the maternal ecology that supports it, that environment is temporally complex, shaped by events and processes that transcend individual agency and precede an individual lifecourse [12, 62]. In their essay "Society: Don't blame the mothers", Richardson et al. (2014) note that through an uncritical DOHaD framing, "[a] mother's individual influence over a vulnerable fetus is emphasized [60, p. 131]." Our findings here in the marmoset monkey help to construct a necessary critical framework. Even when a pregnant person does everything "right," losses may be suffered as a consequence of bodily or historical inertia that individual choices cannot fully bring to a halt. The marmoset mothers in this study are not engaging in "risky" behavior; they would not benefit from prenatal education or mindfulness classes. And yet there are patterns of who is more likely to lose pregnancies and when those losses may occur. We do not blame the monkeys for

the pregnancy outcomes described here as we recognize those outcomes are outside their locus of control and responsibility. Understanding that pregnancy is not a time-bound event but instead embedded in endless temporal, spatial, relational, and social complexity will move us past narratives of blame and help parents and communities better cope with loss.

## Conclusion

In a sample of well-characterized adult female marmosets, stillbirth and early neonatal mortality followed different routes. We did not find evidence for an elevated rate of stillbirth or early neonatal death with maternal weight or increasing maternal weight or age during pregnancy. The path to neonatal death was different, such that maternal early life characteristics of lower birth weight and larger litter size were strongly predictive of those postnatal losses. Our findings suggest that different aspects of an individual pregnancy and the maternal lifecourse underlie mechanisms that differentiate stillbirth and neonatal loss. Conflating stillbirth and early neonatal death under a single umbrella of "perinatal loss" may obscure the particular fragility of early neonatal dependence and attendant needs for parental support as well as potential screening and diagnostic tools that take into account the full arc of a mother's lived experience, from womb to womb and beyond.

## Acknowledgments

We thank the animal care providers, technicians, and veterinarians who tirelessly serve the needs of our study animals. JNR wishes to thank Amanda Dettmer, Agustin Fuentes, Katie Hinde, Robin Nelson, and Rick Smith for providing their invaluable insight on earlier versions of these ideas and this manuscript; and Adelaide Caledonia Goehl for embodying and complicating her experience of motherhood and intergenerationality.

## Author Contributions

**Conceptualization:** Julienne N. Rutherford, Corinna N. Ross, Toni Ziegler, Suzette D. Tardif.

**Data curation:** Julienne N. Rutherford, Toni Ziegler, Victoria A. deMartelly, Laren R. Narapareddy.

**Formal analysis:** Julienne N. Rutherford, Larisa A. Burke, Alana D. Steffen, Laren R. Narapareddy.

**Funding acquisition:** Julienne N. Rutherford, Corinna N. Ross, Toni Ziegler, Suzette D. Tardif.

**Investigation:** Julienne N. Rutherford, Corinna N. Ross, Aubrey Sills, Donna Layne Colon, Suzette D. Tardif.

**Methodology:** Julienne N. Rutherford, Corinna N. Ross, Aubrey Sills, Donna Layne Colon, Suzette D. Tardif.

**Project administration:** Julienne N. Rutherford, Corinna N. Ross, Aubrey Sills, Donna Layne Colon, Laren R. Narapareddy, Suzette D. Tardif.

**Resources:** Toni Ziegler.

**Writing – original draft:** Julienne N. Rutherford.

**Writing – review & editing:** Julienne N. Rutherford, Corinna N. Ross, Toni Ziegler, Larisa A. Burke, Alana D. Steffen, Aubrey Sills, Donna Layne Colon, Victoria A. deMartelly, Laren R. Narapareddy, Suzette D. Tardif.

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
