## [Decision Letter · Decision Letter 0]

18 Feb 2021

PONE-D-21-01098

Maternal litter size and birth weight predict early neonatal death of offspring in the common marmoset monkey

PLOS ONE

Dear Dr. Rutherford,

Thank you for submitting your manuscript to PLOS ONE. After careful consideration, we feel that it has merit but does not fully meet PLOS ONE’s publication criteria as it currently stands. Therefore, we invite you to submit a revised version of the manuscript that addresses the points raised during the review process.

We would ask you to reply to all remarks made by the reviewers in your revised manuscript.

We look forward to receiving your revised manuscript.

Kind regards,

Umberto Simeoni

Academic Editor

PLOS ONE

Journal Requirements:

2)  In your Methods, please provide full details of housing and environmental enrichment.

3) Thank you for including your ethics statement:  "All animal procedures, husbandry, and housing were conducted according to Southwest National Primate Research Center (#1385-CJ-5) and University of Texas Health Sciences Center (#13088x) Institutional Animal Care and Use Committees requirements, certified by the SNPRC and the University of Illinois at Chicago Animal Care Committee (#16-026).".   

Please amend your current ethics statement to confirm that your named ethics committee specifically approved this study.

For additional information about PLOS ONE submissions requirements for ethics oversight of animal work, please refer to http://journals.plos.org/plosone/s/submission-guidelines#loc-animal-research  

4) We note that you have stated that you will provide repository information for your data at acceptance. Should your manuscript be accepted for publication, we will hold it until you provide the relevant accession numbers or DOIs necessary to access your data. If you wish to make changes to your Data Availability statement, please describe these changes in your cover letter and we will update your Data Availability statement to reflect the information you provide.

Reviewers' comments:

Reviewer's Responses to Questions

**Comments to the Author**

1. Is the manuscript technically sound, and do the data support the conclusions?

Reviewer #1: Yes

Reviewer #2: Yes

2. Has the statistical analysis been performed appropriately and rigorously? 

Reviewer #1: Yes

Reviewer #2: Yes

3. Have the authors made all data underlying the findings in their manuscript fully available?

Reviewer #1: Yes

Reviewer #2: Yes

4. Is the manuscript presented in an intelligible fashion and written in standard English?

Reviewer #1: Yes

Reviewer #2: Yes

5. Review Comments to the Author

Reviewer #1: In this review, the authors assess predictors of stillbirth and infant mortality in common marmosets, a neotropical primate species that in captivity regularly produces both twin and triplet litters. The rationale is that the risk factors and mechanisms for stillbirth and early neonatal death may be different and separating them should be informative.

Marmosets are a really cool species and the data in this paper are very valuable to the many researchers seeking to use them for genetic manipulation. What is a little less clear, at least as the paper is currently written, is why they are an “ideal nonhuman primate model” for the questions being studied here. Put bluntly, marmosets are kind of weird and so is their reproductive system, so it is not at all clear why they are a better model for human perinatal mortality than a rhesus or cynomolgus macaque. I think that the authors need to either connect the dots for the reader better, or reframe the study with reference to its value in other arenas. Based on the intro, I was also primed to expect that maternal obesity would be a relevant predictor variable in this study, but did not find out until the discussion that only two of the study females were considered obese.

It is not clear whether the correlations were Pearson’s or Spearman’s correlations and if any corrections for multiple comparisons were used.

I recognize what the authors are trying to say in the conclusions but I would recommend removing the last two sentences.

Reviewer #2: In their manuscript “Maternal litter size and birth weight predict early neonatal death of offspring in the common marmoset monkey” the authors refine and extend their ongoing body of work on intergenerational reproductive programming in marmoset monkeys. This paper importantly differentiates stillbirth and neonatal mortality and the predictors for these outcomes, thereby shining a light on different pathways and mechanisms for these outcomes. This research was conducted in marmosets, a particularly interesting animal system for understanding adaptive variation in reproductive investment as a function of maternal characteristics, with important translational implications for human health relative to other typical biomedical models like rodents and agricultural taxa. The methods are appropriate for the purpose at hand and described in detail. The paper is well-written and will be of broad interest to evolutionary biologists, anthropologists, psychologists, neonatologists, OB/GYN & midwifery clinicians, and others working among the diverse disciplines in developmental programming of health across the life course and inter-generationally.

Given the reviewer instructions for articles in PLOS One to be focused on methodological evaluation, I seek one clarification that caused me confusion in the MS that should be addressed before publication.

Discrepancy between Table 2 & Table 5: Table 2 indicates that N=43 for early neonatal mortality, but Table 5 showing the N for post-natal mortality day only has info for N=33. Table 5 starts at day 1- is that the day following the night of birth? Why are there 10 infants who died in the early neonatal period but don’t have a day post-natal day of death?

For the authors to consider: I additionally provide some comments about interpretation and contextualization that may strengthen the paper for their intended audience.

The authors posit that constraints during their own development, including the parenting behavioral care/milk provisioning triplets experienced may contribute to deficits in their own maternal capacities in the neonatal period and underlie greater infant loss in the later neonatal period (days 3-7). As I read this interpretation in the discussion, I found myself wondering about an issue that wasn’t reported (and the data may not be available, but consideration of it may strengthen the discussion): constraints and sibling exclusion. In birds and mammals, extrinsically or intrinsically constrained parents are delivering inadequate nutrition to support all offspring, but only under the most catastrophic conditions does this result in total litter loss. Rather small differences at birth/hatching among siblings affect dynamics such that healthier, more robust young are able to either elicit more resources from the parent or competitively exclude/inhibit sibling access to parental resources. Based on my reading of the sample description and Table 2, there were N=116 live infant full-term births of which N=43 died in the first 7 post-natal days, for litters in which there were mosaic outcomes (1 or more infant(s) survived while 1 or more infant(s) died) how do the weight trajectories look? Figure 3 is <chef’s kiss=""> and got me thinking about the “did survive” bars and wondering what the weight gain was for survivors whose siblings died vs. survivors in “all surviving” litters (do all surviving litters even happen?). Right now the anchoring/argument in the manuscript is about diminished parenting- but it’s from the perspective of the individual offspring who dies in the neonatal period seemingly from reduced investment. BUT it could be that these triplet females are GREAT parents producing awesome milk and lavishing care, but that these parental efforts only reach a subset of the litter- the ones who survive. I would think that this expands the frame from developmental plasticity and trajectories set in early life to the capacities for substantial flexibility and greater variance across offspring within a litter.

I think the paper would be strengthened by expanding the discussion of implications for public health messaging of causes/sources of pregnancy outcomes in humans. The authors allude to these issues briefly in the conclusion paragraph: “Our findings interrogate mother-blaming narratives of pregnancy loss and other adverse outcomes. The marmosets in this study are not engaging in “risky” behavior.” Many human populations are characterized by cultural beliefs or conventional wisdom that ascribes blame and fault to pregnant women for poor pregnancy outcomes in ways that can be detrimental to women’s mental health and well-being, especially in the wake of infant loss. A paragraph addressing how the present findings, among others, dispel and disrupt women-blaming narratives could improve support for families in these difficult positions and provide clinicians, grief counselors, and religious/spiritual advisors important talking points for those in their care and demonstrate the translational importance of basic science and biomedical model systems.

Minor comments/recommendations for the discussion:

‘90th percentile’ instead of ‘90th%tile’

use 140% or 1.4x, the combination is confusing as currently written: ‘the rate of stillbirth for triplet-born mothers was 140% times greater than for twin-born mothers.”

This sentence gave me pause: “In sum, marmoset mothers of triplets and smaller infants appear to invest less positive parenting behavioral capital in their offspring.” The application of the ‘capital’ construct to parental behavior is an atypical within the life history theory literature, as the capital vs. income construct was predicated on finite energetic resources available to parents for allocation to offspring. Behavior, though supported by the energy that fuels the behavior, is ephemeral, experiential phenomena that can not be “stored” by the infant or “recovered” by the parent. Rather the point being made by the authors seems to be that conditions during early life, including parental behavior, endows the offspring with (among other things) in a trajectory for capacities for parental care. While capital is very specifically about concrete resources, endow includes qualities, abilities, and/or assets. I think the use of the capital construct here should be more explicitly anchored to life history literature and unpacked to justify inclusion, or the point intended could be made without invoking “capital.”</chef’s>

6. PLOS authors have the option to publish the peer review history of their article (what does this mean?). If published, this will include your full peer review and any attached files.

Reviewer #1: No

Reviewer #2: No

---

## [Author Response · Author response to Decision Letter 0]

23 Mar 2021

To the reviewers: 

I am writing to resubmit our manuscript, “Maternal litter size and birth weight predict early neonatal death of offspring in the common marmoset monkey.” We were delighted to receive the reviewers thoughtful and encouraging comments and feel the revised paper has been significantly improved by their suggestions. I have prepared a detailed response letter, including all of our changes and their locations in the text of the marked version of the revised manuscript. (Please also see the attached response letter as it may be easier to read.) We are thankful for the opportunity to be considered for publication in PLOS ONE. 

Most sincerely,

Julienne Rutherford

Editorial requirements 

In your Methods, please provide full details of housing and environmental enrichment. “Animals were housed in family groups in large metal wire enclosures, approximately one meter deep, two meters wide, and one meter tall. Enrichment was provided in the form of hanging and climbing structures and a nest box.” page 7/117-119

Thank you for including your ethics statement: "All animal procedures, husbandry, and housing were conducted according to Southwest National Primate Research Center (#1385-CJ-5) and University of Texas Health Sciences Center (#13088x) Institutional Animal Care and Use Committees requirements, certified by the SNPRC and the University of Illinois at Chicago Animal Care Committee (#16-026).". 

Please amend your current ethics statement to confirm that your named ethics committee specifically approved this study. “All animal procedures, husbandry, and housing were approved by the Southwest National Primate Research Center (SNPRC) Institutional Animal Care and Use Committee requirements, certified by the SNPRC and the University of Illinois at Chicago Animal Care Committee page 7/108

REVIEWER #1

Marmosets are a really cool species and the data in this paper are very valuable to the many researchers seeking to use them for genetic manipulation. What is a little less clear, at least as the paper is currently written, is why they are an “ideal nonhuman primate model” for the questions being studied here. Put bluntly, marmosets are kind of weird and so is their reproductive system, so it is not at all clear why they are a better model for human perinatal mortality than a rhesus or cynomolgus macaque. I think that the authors need to either connect the dots for the reader better, or reframe the study with reference to its value in other arenas. 

Marmosets are indeed complicated nonhuman primate species and are not one-to-one analogs of human reproductive physiology, so the word “model” can be unintentionally definitive. We have changed the wording: 

“The marmoset monkey is an ideal nonhuman primate in which to study the consequences of variability in intrauterine environments and the independent effects of a mother’s fetal and adult phenotypes on her pregnancy outcomes.” page 6/83-84

 Based on the intro, I was also primed to expect that maternal obesity would be a relevant predictor variable in this study, but did not find out until the discussion that only two of the study females were considered obese. 

We agree the obesity framing is a red herring so it was reduced and folded into the previous sentence: 

Maternal health is currently modeled as the amalgamation of a woman’s genome, her current diet and lifestyle choices, her current weight or BMI, her current socioeconomic status, and her racial and/or ethnic characteristics, components frequently described as “risk factors.” page 5/72-77

It is not clear whether the correlations were Pearson’s or Spearman’s correlations and if any corrections for multiple comparisons were used. 

We used Pearson’s correlations and did not correct for multiple comparisons. Our goal here was to explore the data for possible correlations among our predictors to construct the Poisson models, rather than to test hypotheses about relationships between outcomes and predictors, so we chose to be less conservative in our correlation analyses.

“To explore relationships among the predictors in order to construct models, Pearson’s pairwise correlations were performed without correction for multiple comparisons.” page 10/180-182

I recognize what the authors are trying to say in the conclusions but I would recommend removing the last two sentences. 

We have elected instead to expand on this point, per the recommendations of Reviewer #2. 25-26/449-473

Reviewer #2

In their manuscript “Maternal litter size and birth weight predict early neonatal death of offspring in the common marmoset monkey” the authors refine and extend their ongoing body of work on intergenerational reproductive programming in marmoset monkeys. This paper importantly differentiates stillbirth and neonatal mortality and the predictors for these outcomes, thereby shining a light on different pathways and mechanisms for these outcomes. This research was conducted in marmosets, a particularly interesting animal system for understanding adaptive variation in reproductive investment as a function of maternal characteristics, with important translational implications for human health relative to other typical biomedical models like rodents and agricultural taxa. The methods are appropriate for the purpose at hand and described in detail. The paper is well-written and will be of broad interest to evolutionary biologists, anthropologists, psychologists, neonatologists, OB/GYN & midwifery clinicians, and others working among the diverse disciplines in developmental programming of health across the life course and inter-generationally. 

Thank you so much for this lovely summary of our paper and its merit. 

Table 2 indicates that N=43 for early neonatal mortality, but Table 5 showing the N for post-natal mortality day only has info for N=33. Table 5 starts at day 1- is that the day following the night of birth? Why are there 10 infants who died in the early neonatal period but don’t have a day post-natal day of death? 

Day 1 is the day following the night of birth.The discrepancy is due to the limited availability of postnatal mortality data at one of the colonies. 

We have included the following clarifications:

“Chi2 was used to compare pregnancy counts per maternal litter size between the colonies and to compare birth status and cause of death for subset of neonates dying within one week in the Barshop colony.”

page 10/190

“We examined the timing and apparent cause of death of a subset of 33 neonates, all from the Barshop colony, who died within the first postnatal week (Table 5).” 

17/290-291

Reviewer #2 added the following as food for thought rather than required changes: 

The authors posit that constraints during their own development, including the parenting behavioral care/milk provisioning triplets experienced may contribute to deficits in their own maternal capacities in the neonatal period and underlie greater infant loss in the later neonatal period (days 3-7). As I read this interpretation in the discussion, I found myself wondering about an issue that wasn’t reported (and the data may not be available, but consideration of it may strengthen the discussion): constraints and sibling exclusion. In birds and mammals, extrinsically or intrinsically constrained parents are delivering inadequate nutrition to support all offspring, but only under the most catastrophic conditions does this result in total litter loss. Rather small differences at birth/hatching among siblings affect dynamics such that healthier, more robust young are able to either elicit more resources from the parent or competitively exclude/inhibit sibling access to parental resources.

General response: 

We appreciate these points very much, and they are very much at front of mind for marmoset researchers. The broader question of “why litters?” is one we have pondered many times over the years. In fact, this program of research was birthed by the hypothesis that a tradeoff between higher litters and higher perinatal mortality was greater reproductive success for triplet-born females in adulthood, one of the outcomes one might expect if surviving triplets were able to “elicit more resources from the parent or competitively exclude/inhibit sibling access to parental resources.” What we have found instead is that they are not themselves better reproducers. Of course, that is only one of many possible metrics. 

Right now the anchoring/argument in the manuscript is about diminished parenting- but it’s from the perspective of the individual offspring who dies in the neonatal period seemingly from reduced investment. BUT it could be that these triplet females are GREAT parents producing awesome milk and lavishing care, but that these parental efforts only reach a subset of the litter- the ones who survive. I would think that this expands the frame from developmental plasticity and trajectories set in early life to the capacities for substantial flexibility and greater variance across offspring within a litter.

It is correct that our focus here is the perspective of the dead neonate. We think the observations we have made in past studies about how marmoset mothers limit carrying and feeding of triplets and lower birth weight infants provide a strong framework for exploring the hypothesis of developmental trajectories of parenting for these kinds of offspring. Within the hypercompetitive triplet situation, the mother provides less resource (i.e., mothers of triplets are not GREAT parents) and the birth condition of the infants affects how they do in this strained resource environment. Our colony does not cull neonates when litters exceed twins. Thus, offspring who are triplets are initially raised in this hypercompetitive pre-weaning environment and a robust literature suggests this early developmental environment may have implications for diminished parenting capacity later in life. We plan to explore more nuanced aspects of parenting behavior/physiology in triplet-born and extreme birth weight mothers – as well as paternal effects – more intentionally in the prospective data on the offspring and planned future studies.

Based on my reading of the sample description and Table 2, there were N=116 live infant full-term births of which N=43 died in the first 7 post-natal days, for litters in which there were mosaic outcomes (1 or more infant(s) survived while 1 or more infant(s) died) how do the weight trajectories look? Figure 3 is and got me thinking about the “did survive” bars and wondering what the weight gain was for survivors whose siblings died vs. survivors in “all surviving” litters (do all surviving litters even happen?). 

We do have several all surviving litters past the first week. However, daily weights are not taken of infants after the first assessment the morning after they are born so the data needed for the suggested analyses is very limited. For example, we report the weights of a total of only 11 surviving infants in our entire dataset, starting at day 6 through week three (up to day 19). We felt that body weights beyond that were of less relevance to understanding the mechanics of loss during that first week. Of that 11, 6 are the siblings of dead neonates but their weights were recorded days after the death of their sibling. In only one case, all three siblings in the litter were weighed the day one of them died (day 7). 

We modified Figure 2 with an inset highlighting this litter and added the following text: 

“In one case, all siblings were weighed when one was found dead on day 7 (Figure 2, inset). The dead sibling had lost 17.36% of its birth weight, while the two living siblings gained 17.28% and 22.48% respectively. page 18/302-304

I think the paper would be strengthened by expanding the discussion of implications for public health messaging of causes/sources of pregnancy outcomes in humans. The authors allude to these issues briefly in the conclusion paragraph: “Our findings interrogate mother-blaming narratives of pregnancy loss and other adverse outcomes. The marmosets in this study are not engaging in “risky” behavior.” Many human populations are characterized by cultural beliefs or conventional wisdom that ascribes blame and fault to pregnant women for poor pregnancy outcomes in ways that can be detrimental to women’s mental health and well-being, especially in the wake of infant loss. A paragraph addressing how the present findings, among others, dispel and disrupt women-blaming narratives could improve support for families in these difficult positions and provide clinicians, grief counselors, and religious/spiritual advisors important talking points for those in their care and demonstrate the translational importance of basic science and biomedical model systems. 

We agree and very much appreciate the invitation to expand these ideas which we have done in a new section, Public health and health equity implications:

“Our findings of early life contributions to adult pregnancy outcomes in the common marmoset disrupt mother-blaming narratives of pregnancy outcomes in humans. These narratives hold that the pregnant person is solely responsible for pregnancy outcomes and the health of their children, independent of socioecological factors, a moralistic framing that has shaped clinical pregnancy management. A neoliberal emphasis on individualized management of health has largely localized the responsibility for “healthy” pregnancies to the pregnant person’s body and its societally-constructed and clinically-defined “flaws” [33]; for example, shame is employed as a strategy to problematize the flaw of fatness during pregnancy [59]. Mother-blaming narratives are often written in racist ink, where erroneous biological notions of “race” – abstracted from the structural forces of racism in action across generations – are identified as an isolated and deterministic risk factor [60, 61]. The Developmental Origins of Health and Disease (DOHaD) posits that intrauterine and early childhood developmental environments have lifelong implications; this approach too has the potential to concentrate blame on pregnant people by reducing them to the receptacle of that environment [62, 63]. While the fetus is of course inextricably linked to the intrauterine environment and the maternal ecology that supports it, that environment is temporally complex, shaped by events and processes that transcend individual agency and precede an individual lifecourse [14, 64]. In their essay “Society: Don’t blame the mothers”, Richardson et al. (2014) note that through an uncritical DOHaD framing, “[a] mother's individual influence over a vulnerable fetus is emphasized [62, p. 131].” Our findings here in the marmoset monkey help to construct a necessary critical framework. Even when a pregnant person does everything “right,” losses may be suffered as a consequence of bodily or historical inertia that individual choices cannot fully bring to a halt. The marmoset mothers in this study are not engaging in “risky” behavior. And yet there are patterns of who is more likely to lose pregnancies and when those losses may occur. We do not blame the monkeys for the pregnancy outcomes described here as we recognize those outcomes are outside their locus of control and responsibility. Understanding that pregnancy is not a time-bound event but instead embedded in endless temporal, spatial, relational, and social complexity will help us move past narratives of blame and help parents and communities better cope with loss.” page 25-26/452-479

Minor comments/recommendations for discussion:

‘90th percentile’ instead of ‘90th%tile’ 

We have made the change. page 20/323

use 140% or 1.4x, the combination is confusing as currently written: ‘the rate of stillbirth for triplet-born mothers was 140% times greater than for twin-born mothers.” 

We have made the change, omitting the word “times” from the sentence. page 20/330

This sentence gave me pause: “In sum, marmoset mothers of triplets and smaller infants appear to invest less positive parenting behavioral capital in their offspring.” The application of the ‘capital’ construct to parental behavior is an atypical within the life history theory literature, as the capital vs. income construct was predicated on finite energetic resources available to parents for allocation to offspring. Behavior, though supported by the energy that fuels the behavior, is ephemeral, experiential phenomena that can not be “stored” by the infant or “recovered” by the parent. Rather the point being made by the authors seems to be that conditions during early life, including parental behavior, endows the offspring with (among other things) in a trajectory for capacities for parental care. While capital is very specifically about concrete resources, endow includes qualities, abilities, and/or assets. I think the use of the capital construct here should be more explicitly anchored to life history literature and unpacked to justify inclusion, or the point intended could be made without invoking “capital.” 

We appreciate this point. A larger discussion could be had about how energy, time, and behavior are potentially inextricable entities, particularly for a small-bodied, litter-bearing primate. However, we have removed the term “capital” and made the following changes: 

“This intergenerational pattern could play out for marmoset triplets. Marmoset mothers, unlike many other litter-bearing mammals, do not increase the total time invested in a litter as the litter increases in size: “the marmoset mother’s tolerance for carrying infants does not increase with increasing infant stimuli present; rather, marmoset mothers will only tolerate a limited amount of time transporting and nursing infants, regardless of litter size.[16, p. 825]” Triplet and lower birth weight infants were carried for shorter periods of time by their mothers; lower birth weight infants were nursed for shorter periods [16]. Mothers directed aggression to triplet and low birth weight infants at a higher rate than to twin and average weight infants. In sum, marmoset mothers of triplets and smaller infants may invest less time parenting their offspring, and that which is invested may be of lesser quality (e.g., more aggressive).” page 22/370-374, 378-379

---

## [Decision Letter · Decision Letter 1]

5 May 2021

PONE-D-21-01098R1

Maternal litter size and birth weight predict early neonatal death of offspring in the common marmoset monkey

PLOS ONE

Dear Dr. Rutherford,

Thank you for submitting your manuscript to PLOS ONE. After careful consideration, we feel that it has merit but does not fully meet PLOS ONE’s publication criteria as it currently stands. Therefore, we invite you to submit a revised version of the manuscript that addresses the points raised during the review process.

We look forward to receiving your revised manuscript.

Kind regards,

Umberto Simeoni

Academic Editor

PLOS ONE

Journal Requirements:

Reviewers' comments:

Reviewer's Responses to Questions

**Comments to the Author**

1. If the authors have adequately addressed your comments raised in a previous round of review and you feel that this manuscript is now acceptable for publication, you may indicate that here to bypass the “Comments to the Author” section, enter your conflict of interest statement in the “Confidential to Editor” section, and submit your "Accept" recommendation.

Reviewer #1: (No Response)

Reviewer #2: All comments have been addressed

2. Is the manuscript technically sound, and do the data support the conclusions?

Reviewer #1: Yes

Reviewer #2: Yes

3. Has the statistical analysis been performed appropriately and rigorously? 

Reviewer #1: Yes

Reviewer #2: Yes

4. Have the authors made all data underlying the findings in their manuscript fully available?

Reviewer #1: No

Reviewer #2: Yes

5. Is the manuscript presented in an intelligible fashion and written in standard English?

Reviewer #1: Yes

Reviewer #2: Yes

6. Review Comments to the Author

Reviewer #1: There are a few typos remaining (below). I also could not check on data availability.

Line 42: should be “higher birth weight” rather than “higher birth weights”

Line 60: “elide” is a pretty non-standard word choice, I would use a word that people don’t have to look up

Line 161: should be “though” not “thought”

Reviewer #2: The authors have responded to the comments made by the manuscript reviewers. I appreciate the authors' thoughtful response to reviewer comments and look forward to this publication and others hinted at in the response to reviewers.

7. PLOS authors have the option to publish the peer review history of their article (what does this mean?). If published, this will include your full peer review and any attached files.

Reviewer #1: No

Reviewer #2: No

---

## [Author Response · Author response to Decision Letter 1]

6 May 2021

Response to reviewers

We are delighted by this second review of our manuscript. Thank you so much for such positive, constructive feedback throughout this process. We made the three changes suggested by Reviewer #1, outlined below: 

Reviewer #1 (line numbers correspond to marked manuscript): 

• Line 42: We changed “higher birth weights” to “higher birth weight.”

• Line 60: We substituted the phrase “omit, combine, or obscure” for “elide or obscure.”

• Line 161 (actually line 261 in the unmarked manuscript, 262 in the marked manuscript): We changed “thought” to “though.” 

We will be making our data publicly available as soon as possible. 

Sincerely,

Julienne Rutherford, on behalf of all of my co-authors

---

## [Editor Report · Decision Letter 2]

10 May 2021

Early life influence on reproductive success in adulthood in the marmoset monkey: Maternal litter size and birth weight but not adult characteristics predict early neonatal death of offspring

PONE-D-21-01098R2

Dear Dr. Rutherford,

We’re pleased to inform you that your manuscript has been judged scientifically suitable for publication and will be formally accepted for publication once it meets all outstanding technical requirements.

Kind regards,

Umberto Simeoni

Academic Editor

PLOS ONE
---

## [Editor Report · Acceptance letter]

17 May 2021

PONE-D-21-01098R2 

Womb to Womb: Maternal litter size and birth weight but not adult characteristics predict early neonatal death of offspring in the common marmoset monkey 

Dear Dr. Rutherford:

I'm pleased to inform you that your manuscript has been deemed suitable for publication in PLOS ONE. Congratulations! Your manuscript is now with our production department. 

Kind regards, 

on behalf of

Dr. Umberto Simeoni 

Academic Editor

PLOS ONE